# Clinical Outcomes and Oral Health-Related Quality of Life after Periodontal Treatment with Community Health Worker Strategy in Patients with Type 2 Diabetes: A Randomized Controlled Study

**DOI:** 10.3390/ijerph18168371

**Published:** 2021-08-07

**Authors:** Yuan-Jung Hsu, Yi-Hui Chen, Kun-Der Lin, Mei-Yueh Lee, Yu-Li Lee, Chih-Kai Yu, Yuji Kabasawa, Hsiao-Ling Huang

**Affiliations:** 1School of Dentistry, College of Dental Medicine, Kaohsiung Medical University, Kaohsiung 80708, Taiwan; rofatin12345@hotmail.com; 2Dental Department, Kaohsiung Medical University Hospital, Kaohsiung 80756, Taiwan; zakii2710@gmail.com; 3Department of Medicine, College of Medicine, Kaohsiung Medical University, Kaohsiung 80708, Taiwan; berg.kmu@gmail.com; 4Division of Endocrinology and Metabolism, Department of Internal Medicine, Kaohsiung Medical University Hospital, Kaohsiung 80756, Taiwan; lovellelee@hotmail.com; 5Department of Internal Medicine, Kaohsiung Municipal Ta-Tung Hospital, Kaohsiung Medical University Hospital, Kaohsiung 80145, Taiwan; 6Division of Endocrinology and Metabolism, Kaohsiung Municipal Hsiao-Kang Hospital, Kaohsiung 81267, Taiwan; 7Division of Endocrinology and Metabolism, Cishan Hospital, Kaohsiung 84247, Taiwan; jangirl1112@yahoo.com.tw; 8Dental Department, Kaohsiung Municipal Hsiao-Kang Hospital, Kaohsiung 81267, Taiwan; recklessmad@yahoo.com.tw; 9Oral Care for Systemic Health Support, Faculty of Dentistry, School of Oral Health Care Sciences, Graduate School, Tokyo Medical and Dental University, Tokyo 113-8510, Japan; yujikabasawa@gmail.com; 10Department of Oral Hygiene, Kaohsiung Medical University, Kaohsiung 80708, Taiwan

**Keywords:** community health workers, type 2 diabetes mellitus, periodontal disease, quality of life

## Abstract

Interventions engaging community health workers (CHW) for diabetes management aim to improve diabetes care and self-management behaviors among patients. We evaluated the effects of nonsurgical periodontal treatment (NSPT) with the CHW strategy on oral self-care behaviors, periodontal status and oral health-related quality of life (OHQoL) in patients with type 2 diabetes mellitus (T2DM). The participants were randomly assigned to experimental (EG; *n* = 35) and control (CG; *n* = 33) groups. All participants received NSPT, whereas the patients in the EG also received one-on-one 30 min lessons from a CHW over 4 weeks. The EG exhibited greater improvement in the probing pocket depth (β = −0.2, effect size [ES] = 0.61) and clinical attachment level (β = −0.2, ES = 0.59) at 1-month follow-up than the CG did. The ES increased over the 1-, 3- and 6-month follow-ups, indicating an increase in OHQoL (ES = 0.19, 0.60, and 0.62, respectively) in the EG. The patients in the EG were more likely to change their oral self-care behaviors than patients in the CG were. The NSPT with CHW strategy had a positive effect on 1-month periodontal treatment outcomes, long-term OHQoL and oral self-care behaviors in patients with T2DM.

## 1. Introduction

Diabetes has become one of the most pressing health issues worldwide. In Taiwan, the prevalence of diabetes among patients aged 20–79 years was approximately 10.1% in 2014, and it has increased annually [1]. The possible mechanism underlying the interaction between diabetes and periodontal disease has been assessed [2,3,4]. For patients with type 2 diabetes mellitus (T2DM), especially those with long-term poor glycemic control, dentists should assess oral conditions and glycemic control status and treat infections. The use of nonsurgical periodontal treatment (NSPT) to address periodontal disease in patients with T2DM has been frequently discussed in clinical-based studies, which have indicated that NSPT can improve periodontal status and blood sugar control in patients with diabetes [5,6,7,8,9,10]. To improve the treatment of periodontal disease, the control of dental plaque and calculus is vital. A case-control study conducted in Taiwan revealed that interdental cleaning behavior was possibly related to diabetes control status; however, only 32% of patients with diabetes used appropriate tooth-brushing methods, and more than 80% of patients lacked interdental cleaning behavior [11]. Moreover, patients with diabetes have insufficient oral health-related knowledge [12,13]. If patients are unaware of the influence of diabetes on oral health status and the resulting oral complications, they are unable to access appropriate preventive oral healthcare services.

Unlike health educators, known as health education specialists, community health workers (CHWs) are lay people who have in-depth knowledge of the communities they serve, and they usually provide a link between the community and healthcare professionals. The advantages of the CHW strategy include its ability to make up for the shortage of health care professionals in rural communities, and increase the acceptance of patients toward health care or health education. CHWs have a similar culture, lifestyle and language to those of residents; therefore, a wide range of work may be implemented by CHWs [14,15]. The CHW strategy has been effectively employed in response to many health problems, as indicated by its ability to increase the self-examination rate and level of self-efficacy in oral cancer screenings [16], improve diabetes management [17] and improve hypertension management, cardiovascular disease knowledge and self-efficacy [18]. CHWs have also been used to provide education, support and advocacy in several studies related to T2DM self-management [19]. Interventions engaging CHWs in diabetes management have been typically implemented in underserved communities and can improve health, reduce health disparities and enhance health equity [20,21,22].

The American Diabetes Association recommend that patients with diabetes should take good care of their teeth and gums and should go for regular checkups every 6 months [23]. A study found that over 70% of certified diabetes educator nurses did not provide periodontal care for diabetes patients [24], indicating the need for regular periodontal management. Patients with diabetes with periodontal disease do not gain the maximum benefit from treatment because of poor compliance with oral hygiene practices. Therefore, the use of the CHW strategy can provide material conditions and psychosocial support [25] and can improve patients’ compliance on health care behaviors at home. No previous study has focused on the use of the CHW strategy to address periodontal care issues; such a strategy might improve patients’ daily oral self-care behaviors at home, leading to improved treatment outcomes. We hypothesize that NSPT with the CHW strategy results in better treatment outcomes than does NSPT alone. Therefore, we implemented NSPT with the CHW strategy to evaluate outcomes pertaining to periodontal treatment and oral health-related quality of life (OHQoL). Oral health-related knowledge, attitude toward periodontal care and self-care behaviors were also assessed in the present study.

## 2. Materials and Methods

### 2.1. Design and Participants

A randomized experimental design was used. This study was approved by the Institutional Review Board of Kaohsiung Medical University Hospital (reference numbers: KMU-IRB-20140291). We obtained informed consent from all participants. The study was also registered on the ISRCTN Registry (reference number: ISRCTN17756516). 

We recruited participants from the endocrinology and metabolism divisions of Kaohsiung Medical University Hospital, Kaohsiung Municipal Ta-Tung Hospital, and Kaohsiung Municipal Hsiao-Kang Hospital, Taiwan, from 2015 to 2018. Patients aged 35–65, having severe periodontal status [26] with at least one probing pocket depth of at least 5 mm and gingival bleeding during examination, and having at least 16 functional teeth with at least four supporting zones (Category A in the Eichner index) were included in the present study. Participants who had undergone periodontal treatment within the last 6 months, routinely used antibiotics or bisphosphonates or had systematic diseases (such as cancer, kidney or liver failure, or heart disease) were excluded.

Participants were recruited according to the predetermined minimal sample size estimated based on a type I error = 0.05, power = 0.90, and effect size (ES) *f* = 0.25 (medium effect in Cohen’s *f* [27]). The total sample size calculated using repeated-measures analysis of variance, with within–between interaction in G*Power 3.1.5 [28], was 30. With an estimated dropout condition, we included at least twice that number in the study. In total, 32 (86.5%) and 32 (82.1%) patients in the EG and CG, respectively, completed the study at all time points.

### 2.2. Randomization and Blinding

We drew random numbers from a paper bag to divide the participants into the experimental group (EG; odd numbers) and control group (CG; even numbers). A double-blind experimental design was used; participants did not know which group they were assigned to, and the dentists providing NSPT were also blinded to the participant grouping.

### 2.3. Outcome Measures

Periodontal data was collected using the hospitals’ routine periodontal chart with an additional chart designed for the study. We used Kendall’s coefficient of concordance to test the interrater reliability of periodontal examination among three dentists prior to the study, and Kendall’s *W* was 0.929. Glycated hemoglobin (HbA1c) was collected by research staff from the electronic medical records system. A questionnaire survey was conducted to measure OHQoL, oral self-care behaviors, oral health-related knowledge and attitude toward periodontal health. The questionnaire was modified from that used in our previous study [11] and was reviewed by a panel of experts to assess its content validity. Each measure was evaluated for reliability and internal consistency. Other information, such as oral hygiene habits (drinking, betel nut use and smoking) and self-perceived oral health status, was also collected.

#### 2.3.1. Periodontal Index

Five periodontal parameters were measured, namely probing pocket depth (PPD), clinical attachment level (CAL), gingival index (GI) [29], plaque index (PI) [30], and bleeding on probing (BOP). PPD, CAL, and BOP were recorded at six sites (mesial (buccal and lingual/palatal), distal (buccal and lingual/palatal), mid-buccal and mid-lingual/palatal) around each tooth except the third molar. GI and PI were recorded at four sites (mesial, distal, lingual and buccal) around six teeth (16, 12, 24, 36, 32 and 44). All recruited patients underwent full periodontal assessment to measure the changes in these clinical parameters related to periodontal disease.

#### 2.3.2. Glycated Hemoglobin

HbA1c reflects average glycaemia trends over the previous 8 to 12 weeks [31]. We collected the patients’ latest HbA1c level to assess the patients’ diabetes control status.

#### 2.3.3. OHQoL

OHQoL was measured using the Taiwanese version of the 14-item Oral Health Impact Profile-14 (OHIP-14T). The OHIP-14T comprises seven domains: functional limitation, physical disability, physical pain, psychological discomfort, psychological disability, social disability and handicap. Participants respond on the following scale: very often = 4, often = 3, occasionally = 2, rarely = 1 and never = 0. Total scores range from 0 to 56; lower scores indicate a better OHQoL. The evaluation period of OHQoL was the year before treatment (baseline) and each follow-up after treatment.

#### 2.3.4. Oral Self-Care Behaviors

Patients were asked questions regarding their tooth brushing method and interdental brush use. Brushing method was coded as 0 (other methods) or 1 (modified Bass brushing technique), and interdental brush use was coded as 0 (no) or 1 (yes).

#### 2.3.5. Oral Health-Related Knowledge

Thirteen statements were used to measure oral health-related knowledge, including “People with diabetes are more likely to have gum disease” and “It’s better to choose the bigger size interdental brush”. Possible responses included true (1), false (0) or unknown (0), with possible scores ranging from 0 to 13; higher scores indicated a greater degree of oral health-related knowledge. The Kuder–Richardson 20 coefficient was 0.74 for this scale.

#### 2.3.6. Attitude toward Periodontal Health

Seven statements were used to measure attitude toward periodontal health, including “The condition of our teeth can influence our dietary intake” and “I think periodontal disease is serious”. Each statement was evaluated using a 5-point Likert scale from 1 (strongly disagree) to 5 (strongly agree), with possible scores ranging from 7 to 35; higher scores reflected more positive attitudes toward periodontal health. Cronbach’s α was 0.72 for this scale.

### 2.4. Covariates

Demographic characteristics encompassing five variables, namely gender, age, duration of diabetes, body mass index (BMI) and education level, were assessed at baseline for each participant in this study.

### 2.5. CHWs Recruitment and Training

Four CHWs were recruited from the community by using flyers and social media. We conducted a 4 h training course containing a curriculum on periodontal disease and care, teaching and communication skills. The CHWs received a training manual reviewed by a panel of experts. The training manual described the goals and contents of each lesson in detail. All four candidates passed the test and became certified CHWs. The research staff maintained contact with CHWs to address any difficulties in reading the training manual during the month prior to the intervention.

### 2.6. Intervention

The T2DM patients in the EG received the NSPT and periodontal care curriculum taught by CHWs. The curriculum included four 30 min, one-on-one lessons taught by CHWs during NSPT. The teaching dates and locations were determined according to the participants’ convenience. To encourage their effort and participation, the CHWs received USD 70 upon completing the curriculum with each patient. The primary goal of the lessons was to provide information in the selection of tooth-brushing tools, correct toothbrushing methods based on the Bass and modified Bass technique and periodontal care related to diabetes mellitus (e.g., the necessity of periodontal treatment and the two-way relationship between diabetes and periodontal disease). Appendix A summarizes the periodontal care curriculum.

Standardized lecture slides were prepared based on the curriculum and presented on a tablet computer. The tablet computers featured the ability to enlarge the slides to help patients understand the course content and a built-in camera function to help patients check their cleaning condition after using a plaque disclosing agent. A set of toothbrushing tools comprising a soft-bristled toothbrush, fluoride toothpaste, dental floss, an interdental brush, a mouthwash cup and fluoride mouthwash was offered to participants to help them learn how to select and use toothbrushing tools. A working log was designed for CHWs to record the implementation of the intervention and the degree of participants’ cooperation. Using the working log, the CHWs and researchers could evaluate the patients’ understanding of the intervention process and the effectiveness of teaching in each course to ensure that the participants achieved the course goals.

The patients with T2DM in the control group (CG) received only NSPT. A simple leaflet presenting information on the relationship between periodontal disease and diabetes was provided to patients in the CG at the end of the study. The NSPT included full-mouth scaling, root planning and oral hygiene instructions (OHI). The course of treatment was completed in 1 to 2 months, depending on patients’ periodontal status. Routine OHI based on personal oral conditions were provided by a dental hygienist, including tooth brushing technique, flossing skills and interdental aids.

### 2.7. Data Collection

Data were collected by trained and certified study personnel at four time points, namely at baseline and at the 1-, 3- and 6-month follow-ups. At baseline, our staff assisted participants in completing the self-administered questionnaire, and patients were then appointed to receive oral examination by dentists to obtain the periodontal parameters. The participants completed an identical questionnaire and oral examination at the 1-, 3- and 6-month follow-up appointments. The HbA1c data were collected at baseline and at the 3- and 6-month follow-ups by staff using an electronic medical records system under the authority of the endocrinology and metabolism doctor.

### 2.8. Statistical Analysis

This study explored the relationships among the variables by using STATA version 13.0 (Stata Corp LP, College Station, TX, USA). A chi-squared test and two-sample *t*-test were used to compare the demographic variables of the EG and CG. A logistic regression model using generalized estimating equations (GEEs) analyzed the change in oral self-care behaviors between baseline and the follow-up appointments. Comparisons of periodontal parameters, OHQoL, HbA1c, knowledge, and attitude between the EG and CG were analyzed with a linear regression model using GEEs. The intervention effects were adjusted for age, gender, educational level, BMI and T2DM duration. The ES (Cohen’s d) of continuous variables were calculated from the mean difference between baseline and the follow-ups within the EG and CG. An effect of 0.20 is small, 0.50 is medium and 0.80 is large [27].

## 3. Results

### 3.1. Recruitment

Figure 1 presents the CONSORT [32] flow chart of patient recruitment for the present randomized controlled trial. The dropout analysis revealed no difference in any of the baseline characteristics at each follow-up between the patients who were included in the analysis and those who dropped out.

### 3.2. Baseline Information between the Two Groups

The distribution of sociodemographic characteristics between the EG and CG are shown in Table 1. No differences were observed between the two groups in any variable.

### 3.3. Intervention Effects on Knowledge, Attitude, and Behaviors

The changes in oral health-related knowledge, attitude toward periodontal health, and oral self-care behaviors at different stages are presented in Table 2. The EG exhibited a greater increase at 1-, 3- and 6-month follow-up with respect to oral health-related knowledge (β = 4.5, 95% confidence interval [CI] = 3.42, 5.54, ES = 1.82; β = 3.4, 95% CI = 2.05, 4.66, ES = 1.36; and β = 3.5, 95% CI = 2.04, 4.89, ES = 1.44, respectively) and attitude toward periodontal health (β = 3.2, 95% CI = 1.58, 4.75, ES = 1.02; β = 3.6, 95% CI = 1.66, 5.45, ES = 1.05; and β = 3.0, 95% CI = 1.01, 5.08, ES = 0.79, respectively) than the CG did. The EG was more likely to use the modified Bass technique at the 1- and 3-month follow-ups (adjusted odds ratio (aOR) = 12.2, 95% CI = 2.31, 64.98; aOR = 11.6, 95% CI = 1.15, 116.65, respectively) and the interdental brush at the 1-month follow-up (aOR = 4.1, 95% CI = 1.02, 16.29) than the CG was.

### 3.4. Intervention Effects on Periodontal Parameters

Table 3 presents the changes in periodontal parameters at different stages by group among the patients with T2DM. The EG had a greater reduction in PPD (β = −0.2, 95% CI = −0.41, −0.08, ES = 0.61) and CAL (β = −0.2, 95% CI = −0.39, −0.06, ES = 0.59) at the 1-month follow-up than the CG did. However, no differences were observed between the EG and CG in GI, PI, or BOP. At the 1-month, 3-month and 6-month follow-ups, medium and large ESs were observed in GI (EG: 1.56, 1.24, and 1.03; CG: 1.05, 1.67, and 1.22, respectively), PI (EG: 1.07, 0.93, and 0.92; CG: 0.98, 1.17, and 1.17, respectively) and BOP (EG: 1.36, 1.28, and 1.13; CG: 0.79, 1.09, and 0.88, respectively) in both groups.

### 3.5. Intervention Effects on OHQoL and HbA1c

Table 4 presents the changes in OHQoL and HbA1c at different stages by group among the patients with T2DM. The CG exhibited a greater improvement than the EG did in OHQoL at the 1-month follow-up (β = 3.2, 95% CI = 0.90, 5.46, ES = 0.56). A medium ES (0.62) was observed at the 1-month follow-up in the CG and at the 3- and 6-month follow-ups in both groups (EG: 0.60 and 0.62; CG: 0.67 and 0.67, respectively). No difference in the reduction of HbA1c levels was observed between the EG and the CG at the follow-ups.

### 3.6. Satisfaction Survey of the Intervention

We converted the 5-point scale of the satisfaction survey into a 100-point scale, as shown in Figure 2. Because the mean scores of all items were greater than 90 points, the results of the satisfaction survey indicate that the patients were satisfied with the periodontal care intervention.

## 4. Discussion

This is the first study employing the CHW strategy to improve daily oral self-care behaviors in patients with T2DM, with the aim of improving treatment outcomes.

We observed a greater improvement in 1-month short-term clinical outcomes with respect to PPD and CAL in the EG than in the CG. Using the CHW strategy in the EG improved patients’ toothbrushing technique and interdental brush use in home care at the 1-month follow-up, which may contribute to improved clinical outcomes after periodontal treatment. Moreover, the positive changes in oral health-related knowledge and attitude toward periodontal health at all follow-ups were higher in the EG than in the CG.

In the present study, no differences were found between EG and CG in OHQoL at 3-month and 6-month follow-ups. However, both EG and CG exhibited improvement in OHQoL at 3-month and 6-month follow-ups. An increase in ES in OHQoL from 1-month to 6-month follow-ups was observed in the EG and the ES was medium at 3-month and 6-month follow-ups. This positive effect of using CHWs is consistent with the findings of previous studies focused on different health issues [17,18,19]. The medium ES in OHQoL was also observed in CG at 1-month, 3-month and 6-month follow-ups, which is consistent with the findings of previous studies that routine nonsurgical therapy can moderately improve OHQoL in adults with periodontal disease [33,34,35]. Therefore, after patients receive periodontal treatment, their quality of life may improve in the long term regardless of whether the CHW strategy is implemented. Notably, the improvement in OHQoL was greater in the CG than in the EG at 1-month follow-up. We speculate that after receiving periodontal care intervention, the patients in the EG had a clear improvement in their knowledge and attitudes toward periodontal care, which may have indirectly increased their attention toward oral health conditions and susceptibility to the influence of periodontal disease on their lives, and this may require more time to exhibit long-term effects on OHQoL.

A greater improvement in PPD and CAL was observed in the EG than in the CG at the 1-month follow-up. Relatively large ESs in PPD, CAL and BOP were also noted in the EG than in the CG. This observation may have occurred because the health education provided by diabetes educators to patients does not include adequate content related to periodontal care; one previous study even indicated that most education curricula from diabetes educators do not include an oral health module [36]. Although conventional NSPT providers administered the OHI during the treatment course, the dental hygienist might give the same OHI for patients with different levels of risk of periodontal disease and may not give instructions to patients at every treatment [37], which may result in insufficient periodontal self-care in T2DM patients. One study found a correlation between oral health behavior and calculus accumulation [38]. Calculus is a contributing factor to periodontal diseases; therefore, the periodontal care curriculum taught by CHWs in our study emphasized the relationship between diabetes and periodontal disease to improve patients’ self-care behavior at home, which may contribute to short-term treatment outcomes.

In this study, the patients in the EG demonstrated good effects on changes in knowledge, attitude and behaviors. This finding is consistent with previous studies using CHWs for other health issues [16,17,18]. Previous studies have indicated that good health care outcomes depend on patients’ adherence to recommended treatment regimens; furthermore, the key to ensuring patient adherence is a comprehensive understanding of patients’ attitudes, cultural context, social support systems and challenges to emotional health [39,40]. Therefore, the positive changes in patients’ oral healthcare behaviors may be due to the empathetic and skilled CHWs who built close relationships with patients and alleviated their stress of learning, which increased patient adherence. Moreover, the content of the periodontal care curriculum provided clear and essential knowledge on NSPT and oral self-care techniques that clinical OHI may not provide to every patient, as suggested by the high scores in the satisfaction survey. Notably, the number of patients using interdental brushes in the EG decreased at the 6-month follow-up, which suggests that, in future studies, CHWs can provide phone-based education or reminders during follow-up to ensure that patients continue practicing self-care behavior.

In our study, the average HbA1c level of neither group showed clinical improvement at follow-up after patients received NSPT; this finding is consistent with that of a previous study [41]. According to a systematic review, evidence is insufficient for supporting that periodontal treatment improves glycemic control, and that the favorable effects can be maintained after 4 months [42]. The reason why improvement in glycemic control was not achieved in our study may be because the patients’ blood glucose levels at baseline were already well controlled to begin with (average HbA1c level < 7%, according to the definition of American Diabetes Association [23]). Furthermore, a systematic review indicated that the effect of periodontal treatment on glycemic control was much more obvious in studies with a higher baseline HbA1c level than in those with a lower baseline HbA1c level [43]. Another previous study indicated that the effect of lowering glucose differs based on factors such as the intervention method, duration of diabetes, baseline glycemia and whether the patient had undergone previous therapy [44].

The present study has several limitations. First, the attrition rate of the entire study was 15.8% (EG: 13.5% and CG: 17.9%, respectively) from baseline to 6-month follow-up; however, a dropout analysis using baseline data indicated no difference in demographic characteristics between dropouts and participants. Second, maturation bias may have occurred because the teaching skill of the CHWs may have improved over time. Patients who received curriculum later may receive more proficient teaching. However, this bias was potentially limited by the standardized design of our curriculum. Third, social desirability may influence results, especially in the EG. Patients might provide responses that conform to social expectations rather than responses that accurately reflect their condition. Fourth, the patients were not placed under any special restrictions regarding daily diet, drug use or treatment methods related to their diabetes during the research period; these factors should be carefully controlled in future research. Finally, the participants were patients in a hospital; therefore, the generalization of findings to other T2DM patients should be carefully discussed.

## 5. Conclusions

This study reported that NSPT with the CHW strategy improved short-term clinical treatment outcomes (i.e., PPD and CAL) and exerted a positive effect on long-term OHQoL in T2DM patients. The CHW strategy also had a positive effect on oral health-related knowledge, attitude toward periodontal care and oral self-care behaviors. Our study further suggests that in areas where health care professionals are scarce, CHWs should be employed to improve preventive oral (specifically, periodontal) self-care behavior as part of diabetes management.

## Figures and Tables

**Figure 1 ijerph-18-08371-f001:**
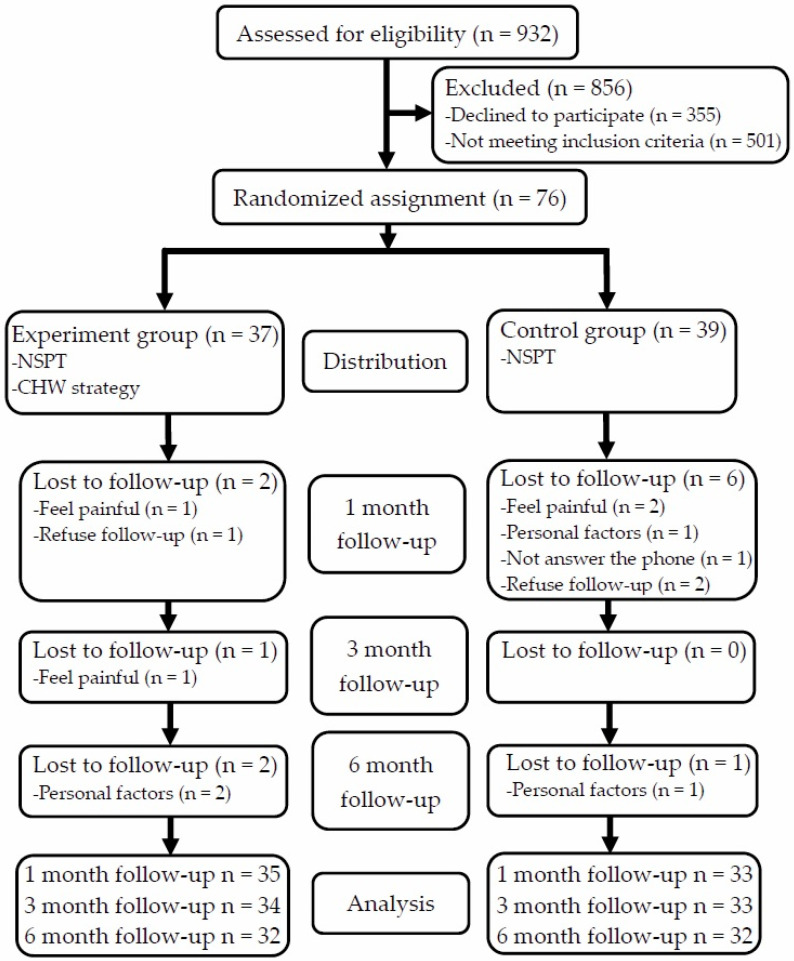
CONSORT flow chart of participant recruitment.

**Figure 2 ijerph-18-08371-f002:**
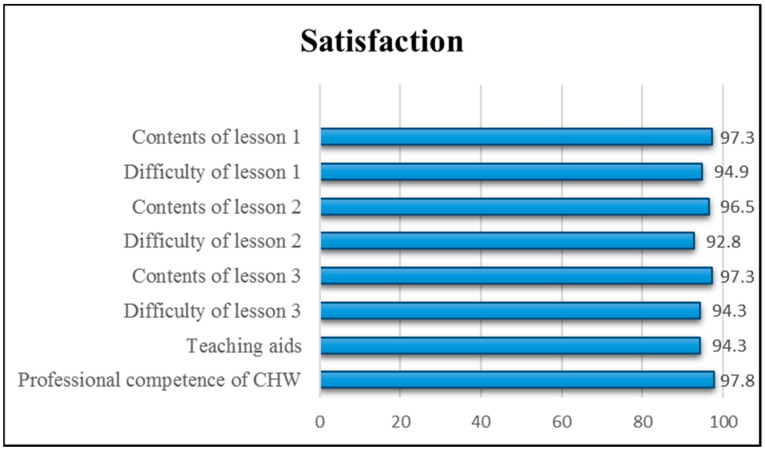
Satisfaction survey results of EG on periodontal care intervention (N = 35).

**Table 1 ijerph-18-08371-t001:** Distribution of sociodemographic characteristics in patients with T2DM.

	EG (*n* = 35)		CG (*n* = 33)		
	*N*	%	*N*	%	*p*
Gender					
Male	18	(51.4)	22	(66.7)	0.202
Female	17	(48.6)	11	(33.3)	
Age, mean ± SD ^†^	54.7 ± 6.1		54.8 ± 6.9		0.932
Duration of diabetes (year)					
<5	16	48.5	12	36.4	0.273
5–10	10	30.3	8	24.2	
>10	7	21.2	13	39.4	
Baseline HbA1c(%), mean ± SD ^†^	6.9 ± 1.3		6.9 ± 1.2		0.969
BMI (kg/m^2^), mean ± SD ^†^	27.1 ± 4.5		26.9 ± 4.2		0.826
Education level					
Less than junior high school	7	20.0	6	18.2	0.976
High school	17	48.6	16	48.5	
College and above	11	31.4	11	33.3	
Drinking					
Yes	5	14.3	6	18.2	0.663
No	30	85.7	27	81.8	
Betel nut chewing					
Yes	2	5.7	1	3.0	0.590
No	33	97.3	32	97.0	
Smoking					
Yes	5	14.3	6	18.2	0.663
No	30	85.7	27	81.8	
Self-perceived oral health status					
Very good/Good	8	22.9	4	12.1	0.172
Common	20	57.1	16	48.5	
Very poor/Poor	7	20.0	13	39.4	

SD: standard deviation. ^†^ Two-sample *t*-test results (other values are from a chi-squared test).

**Table 2 ijerph-18-08371-t002:** Effect of changes in oral health-related knowledge, attitude toward periodontal health, and oral self-care behaviors at different stages (at baseline and at 1-month, 3-month and 6-month follow-up) by group among patients with T2DM.

	EG (*n* = 35)		CG (*n* = 33)				
	Diff ± SD ^†^		Diff ± SD ^†^		Effect Size ^d^	β/aOR	(95% CI)
Oral health-related knowledge (0–13)							
Group (EG) × Time (second)	5.6 ± 2.8		0.9 ± 2.4		1.82	4.5	(3.42, 5.54)
Group (EG) × Time (third)	5.8 ± 2.8		2.2 ± 2.5		1.36	3.4	(2.05, 4.66)
Group (EG) × Time (fourth)	6.1 ± 2.7		2.4 ± 2.4		1.44	3.5	(2.04, 4.89)
Effect size ^a^	2.02		0.39				
Effect size ^b^	2.09		0.85				
Effect size ^c^	2.25		0.98				
Attitude toward periodontal health (7–35)							
Group (EG) × Time (second)	4.8 ± 3.5		1.5 ± 2.9		1.02	3.2	(1.58, 4.75)
Group (EG) × Time (third)	5.4 ± 3.9		1.7 ± 3.0		1.05	3.6	(1.66, 5.45)
Group (EG) × Time (fourth)	5.7 ± 3.7		2.8 ± 3.8		0.79	3.0	(1.01, 5.08)
Effect size ^a^	1.40		0.52				
Effect size ^b^	1.39		0.57				
Effect size ^c^	1.55		0.73				
Oral self–care behaviors							
Use modified Bass brushing technique (*n*, %) ^‡^							
Group (EG) × Time (second)	23	65.7	7	21.2		12.2	(2.31, 64.98)
Group (EG) × Time (third)	23	67.7	15	45.5		11.6	(1.15, 116.64)
Group (EG) × Time (fourth)	20	62.5	13	40.6		7.2	(0.98, 53.32)
Use interdental brush (*n*, %) ^‡^							
Group (EG) × Time (second)	19	54.3	8	24.2		4.1	(1.02, 16.29)
Group (EG) × Time (third)	18	52.9	11	33.3		1.9	(0.36, 9.90)
Group (EG) × Time (fourth)	14	43.8	10	31.3		1.2	(0.20, 6.98)

Time (second): 1-month follow-up; Time (third): 3-month follow-up; Time (fourth): 6-month follow-up; SD: standard deviation; CI: confidence interval. ^†^ Difference between baseline and follow-up within group. ^‡^ Number of positive behavioral changes between baseline and follow-up within group. ^a^ Effect size calculated as mean difference between baseline and 1-month follow-up. ^b^ Effect size calculated as mean difference between baseline and 3-month follow-up. ^c^ Effect size calculated as mean difference between baseline and 6-month follow-up. ^d^ Effect size calculated as mean difference of change between baseline and follow-up measurements between EG and CG. β/aOR (95% CI): adjusted for age, gender, education level, BMI, and duration of diabetes in linear or logistic regression using generalized estimating equations (GEEs); Reference: Control group × Time (baseline).

**Table 3 ijerph-18-08371-t003:** Effect of changes in periodontal parameters at different stages (at baseline and at 1-month, 3-month and 6-month follow-up) by group among patients with T2DM.

	EG (*n* = 35)	CG (*n* = 33)			
	Diff ± SD ^†^	Diff ± SD ^†^	Effect Size ^d^	β	(95% CI)
PPD					
Group (EG) × Time (second)	−0.5 ± 0.4	−0.2 ± 0.3	0.61	−0.2	(−0.41, −0.08)
Group (EG) × Time (third)	−0.5 ± 0.5	−0.4 ± 0.4	0.22	−0.1	(−0.34, 0.07)
Group (EG) × Time (fourth)	−0.4 ± 0.5	−0.3 ± 0.5	0.18	−0.1	(−0.36, 0.11)
Effect size ^a^	1.19	0.73			
Effect size ^b^	0.94	0.85			
Effect size ^c^	0.80	0.67			
CAL					
Group (EG) × Time (second)	−0.4 ± 0.4	−0.1 ± 0.3	0.59	−0.2	(−0.39, −0.06)
Group (EG) × Time (third)	−0.3 ± 0.4	−0.2 ± 0.4	0.14	−0.1	(−0.30, 0.14)
Group (EG) × Time (fourth)	−0.3 ± 0.5	−0.2 ± 0.5	0.15	−0.1	(−0.37, 0.16)
Effect size ^a^	0.89	0.41			
Effect size ^b^	0.69	0.55			
Effect size ^c^	0.59	0.43			
GI					
Group (EG) × Time (second)	−0.8 ± 0.5	−0.8 ± 0.8	0.01	0.0	(−0.27, 0.30)
Group (EG) × Time (third)	−0.9 ± 0.7	−1.1 ± 0.6	0.30	0.2	(−0.16, 0.50)
Group (EG) × Time (fourth)	−0.8 ± 0.8	−1.0 ± 0.8	0.22	0.2	(−0.17, 0.51)
Effect size ^a^	1.56	1.05			
Effect size ^b^	1.24	1.67			
Effect size ^c^	1.03	1.22			
PI					
Group (EG) × Time (second)	−0.6 ± 0.6	−0.8 ± 0.8	0.24	0.2	(−0.12, 0.46)
Group (EG) × Time (third)	−0.7 ± 0.7	−1.0 ± 0.9	0.44	0.3	(−0.04, 0.63)
Group (EG) × Time (fourth)	−0.7 ± 0.7	−1.0 ± 0.8	0.32	0.2	(−0.14, 0.58)
Effect size ^a^	1.07	0.98			
Effect size ^b^	0.93	1.17			
Effect size ^c^	0.92	1.17			
BOP (%)					
Group (EG) × Time (second)	−26.5 ± 19.5	−20.5 ± 26.0	0.26	−5.7	(−15.98, 3.88)
Group (EG) × Time (third)	−29.5 ± 23.0	−28.7 ± 26.4	0.03	−1.6	(−12.66, 9.47)
Group (EG) × Time (fourth)	−30.4 ± 26.9	−26.1 ± 29.8	0.15	−0.3	(−12.04, 11.35)
Effect size ^a^	1.36	0.79			
Effect size ^b^	1.28	1.09			
Effect size ^c^	1.13	0.88			

Time (second): 1-month follow-up; Time (third): 3-month follow-up; Time (fourth): 6-month follow-up; SD: standard deviation; CI: confidence interval. ^†^ Difference between baseline and follow-up within group. ^a^ Effect size calculated as mean difference between baseline and 1-month follow-up. ^b^ Effect size calculated as mean difference between baseline and 3-month follow-up. ^c^ Effect size calculated as mean difference between baseline and 6-month follow-up. ^d^ Effect size calculated as mean difference of change between baseline and follow-up measurements between EG and CG. Β (95% CI): adjusted for age, gender, education level, BMI, and duration of diabetes in linear regression using generalized estimating equations (GEEs); Reference: Control group × Time (baseline).

**Table 4 ijerph-18-08371-t004:** Effect of changes in oral health-related quality of life and HbA1c at different stages (at baseline and at 1-month, 3-month and 6-month follow-up) by group among patients with T2DM.

	EG (*n* = 35)	CG (*n* = 33)			
	Diff ± SD ^†^	Diff ± SD ^†^	Effect Size ^d^	β	(95% CI)
Oral health-related quality of life					
Group (EG) × Time (second)	−0.9 ± 4.5	−4.0 ± 6.5	0.56	3.2	(0.90, 5.46)
Group (EG) × Time (third)	−2.9 ± 4.8	−4.1 ± 6.1	0.22	1.2	(−1.70, 4.07)
Group (EG) × Time (fourth)	−3.2 ± 5.2	−4.3 ± 6.5	0.19	1.0	(−2.29, 4.19)
Effect size ^a^	0.19	0.62			
Effect size ^b^	0.60	0.67			
Effect size ^c^	0.62	0.67			
HbA1c					
Group (EG) × Time (third)	−0.1 ± 1.1	−0.1 ± 0.8	0.02	0.1	(−0.64, 0.83)
Group (EG) × Time (fourth)	0.1 ± 1.1	0.1 ± 0.4	0.04	0.2	(−0.67, 1.07)
Effect size ^b^	0.07	0.12			
Effect size ^c^	0.09	0.16			

Time (second): 1-month follow-up; Time (third): 3-month follow-up; Time (fourth): 6-month follow-up; SD: standard deviation; CI: confidence interval. ^†^ Difference between baseline and follow-up within the group. ^a^ Effect size calculated and as the mean difference between baseline and 1-month follow-up measurement. ^b^ Effect size calculated and as the mean difference between baseline and 3-month follow-up measurement. ^c^ Effect size calculated and as the mean difference between baseline and 6-month follow-up measurement. ^d^ Effect size calculated and as the mean difference of change between baseline and follow-up measurements between the EG and CG. Β (95% CI): adjusted for age, gender, education level, BMI and duration of diabetes by linear regression in Generalized Estimating Equations (GEE); Reference: Control group × Time (baseline).

## Data Availability

To ensure confidentiality, the data sets generated and analyzed in this study are not publicly available, but they may be obtained from the corresponding author (e-mail: hhuang@kmu.edu.tw) upon reasonable request.

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
