# Peer review of "Clinical Outcomes and Oral Health-Related Quality of Life after Periodontal Treatment with Community Health Worker Strategy in Patients with Type 2 Diabetes: A Randomized Controlled Study"

_ijerph, 2021, doi:10.3390/ijerph18168371_

Round 1

Reviewer 1 Report

Dear authors

It is a very important contribution for the readers, the methodology used is impeccable of very high quality and the contributions are excellent. Congratulations

Reviewer 2 Report

Dear Authors,

Thank you for submitting your article to this prestigious journal.

The manuscript is well presented but it presents some issues:

Abstract: Abstract should be revised and make it more appealing.

Keywords: To ensure a properly research in medical databases, use MeSH terms to find  appealing keywords that could be helpful in finding your article.

For patients with type 2 48 diabetes mellitus (T2DM), especially those with long-term poor glycemic control, dentists 49 should assess oral conditions and glycemic control status and treat infections. Using non- 50 surgical periodontal treatment (NSPT) to address periodontal disease in patients with 51 T2DM has been frequently discussed ” More citations are needed.

"CHWs have a similar culture, lifestyle, and lan- 66 guage to those of residents; therefore, a wide range of work may be implemented by 67 CHWs [15,16]. The CHW strategy has been effective in many health issues, including in- 68 creasing self-examination and the self-efficacy of oral cancer screenings [17], improving 69 diabetes management [18], and improving hypertension management and cardiovascular 70 disease knowledge and self-efficacy [19]. ” authors should describe why CHWs these sentences are only referable to them.

Aim should be revised making it more appealing.

Methods and results sections are well presented and written.

Figures and tables are well presented.

“A chi-square test and two sample t test 209 were used to compare the demographic variables of the EG and CG. A logistic regression 210 model using generalized estimating equations (GEEs) analyzed the change in oral self- 211 care behaviors between baseline and the follow-up appointments.” Why for the statistical analysis the authors have chosen the chi-squared instead of the anova?

Notably, the improvement in OHQoL was greater in the CG than in the EG at the 1- 282 month follow-up. In addition, the increasing ES in OHQoL from the 1-month (Time 2) to 283 6-month (Time 4) follow-ups was observed in both groups. Both the EG and the CG 284 achieved within-group medium ES in OHQoL at 3-month and 6-month follow-ups. There- 285 fore, after patients receive periodontal treatment, their quality of life may improve in the 286 long term regardless of whether the CHW strategy is implemented.” Citations are needed to support your findings.

Discussion should be revised.

Please be more specific regarding the limitation.

Conclusion must be entirely revised making it more appealing.

English spell revision is necessary.

Best Regards.

Reviewer 3 Report

Remark - Meticulously described methods, a large amount of results difficult to follow !!! but the discussions are very short.

Recommendation - please, expand the discussions section so as to highlight the results and make their meaning easier to understand.

Reviewer 4 Report

This study is evaluated that the effect of non-surgical periodontal treatment (NSPT) using community health workers (CHW) on periodontal disease status and oral health-related quality of life (OHQoL) in patients with type 2 diabetes. Subject sampling, number, testing methods, and statistical methods were not particularly problematic, and the results showed that CHW intervention subjects were more likely to alter oral self-care behavior than control subjects. The NSPT used CHW strategies was simple and meaningful, with positive effects on 1-month periodontal outcomes, long-term OHQoL, and oral self-care behavior in T2DM patients.

Therefore, this paper is considered to be suitable for publication in  ijerph.
